# The intervertebral disc during growth: Signal intensity changes on magnetic resonance imaging and their relevance to low back pain

Teija Lund[1][ORCID][‡][*], Dietrich Schlenzka[2][ORCID][‡], Martina Lohman[3][ORCID], Leena Ristolainen[2][ORCID], Hannu Kautiainen[4][ORCID], Erkko Klemetti[2][‡], Kalevi Österman[2][‡]

1 Department of Orthopaedics and Traumatology, University of Helsinki and Helsinki University Hospital, Helsinki, Finland, 2 Research Institute Orton, Helsinki, Finland, 3 Department of Radiology, University of Helsinki and Helsinki University Hospital, Helsinki, Finland, 4 Medcare Oy, Espoo, Finland

⊚ These authors contributed equally to this work.
‡ These authors also contributed equally to this work. TL and DS are joint senior authors.
* lundteija@gmail.com

**Data Availability Statement:** All relevant data are within the article and its Supporting Information files.

## Abstract

Life-time prevalence of low back pain (LBP) in children and adolescents varies from 7% to 72%. Disc changes on magnetic resonance imaging (MRI) have been reported in up to 44% of children with earliest observations around pre-puberty. In this longitudinal cohort study, our objective was to determine the natural history of disc changes from childhood to early adulthood, and the possible association of these changes to LBP. Healthy 8-year-old schoolchildren were recruited for this longitudinal study consisting of a semi-structured interview, a clinical examination, and an MRI investigation at the age of 8–9 (Y8), 11–12 (Y12) and 18–19 (Y19) years. The interview inquired about LBP without trauma. T2-weighted sagittal MRI of the lumbar spine was acquired. Life-long prevalence of LBP was determined, and the disc signal intensity (SI) at the three lowest lumbar levels was assessed both visually using the Schneiderman classification *(Bright-Speckled-Dark)*, and digitally using the disc to cerebrospinal fluid -SI ratio. Possible associations between SI changes and LBP were analyzed. Ninety-four of 208 eligible children were included at Y8 in 1994, 13 and 23 participants were lost to follow-up at Y12 and Y19, respectively. Prevalence of LBP increased after the pubertal growth spurt reaching 54% at Y19. On MRI, 18%, 10% and 38% of participants had disc SI changes at Y8, Y12 and Y19, respectively. No significant associations between self-reported LBP and either qualitative or quantitative disc SI changes were observed at any age. Life-time prevalence of LBP reached 54% by early adulthood. Disc SI changes on MRI traditionally labeled as degenerative were seen earlier than previously reported. Changes in disc SI were not associated with the presence of LBP in childhood, adolescence or early adulthood.

**Funding:** This research (official recipient: DS) was supported by The Research Institute of the Invalid Foundation (later Research Institute Orton) through grants by the Ministry of Social Affairs and Health in Finland (Project Identification Number A2500/465), and by the Siviä Kosti Foundation of the Invalid Foundation in Helsinki, Finland. The funders did not play any role in the study design, data collection and analysis, decision to publish, or preparation of the manuscript.

**Competing interests:** The authors have declared that no competing interests exist.

## Introduction

In recent years, low back pain (LBP) in children and adolescents has been recognized as a major public health concern. What was once deemed a rare occurrence is now acknowledged a common condition with possible repercussions into adult life.

Varying prevalence of LBP in children and adolescents has been reported depending on study design, definition of LBP, recall period, and age of study participants. In a systematic literature review, the lifetime prevalence of LBP in adolescence ranged from 7 to 72% [1]. Although most children and adolescents report no LBP or low probability of LBP, 16 to 37% suffer occasional bouts, and up to 10% report repeated episodes [2]. The prevalence of LBP increases with age [1, 3–10] reaching adult levels by the end of puberty [5]. A clear association has been suggested between puberty and back pain [11] at least partly explained by the growth spurt [12]. Childhood LBP seems to be a significant risk factor for LBP in adulthood [13–15] with reported odds ratios from 3.5 to 4 [14, 15].

The gradual loss of water content associated with "aging" of the intervertebral disc is manifested by a reduced signal intensity (SI) of the nucleus pulposus on T2-weighted magnetic resonance imaging (MRI). Several studies have reported an association between disc degeneration (DD) and LBP in adults [16–18]. Although signs of DD on MRI are common in asymptomatic individuals as well [17, 18], a meta-analysis in an adult population concluded that they are more prevalent in the LBP population [19]. In pioneering research on adolescents, about one third of 15-year-olds were found to have at least one degenerated disc on MRI [20, 21]. While no significant difference in the prevalence of MRI findings between adolescents with or without LBP was noticed in the earliest studies [20], those with DD at an early age seemed to be at greater risk of having recurrent LBP in the future [21, 22]. In a more recent study, 35% of 13- to 20-year-old subjects presented with DD on MRI, and almost one third of them had multilevel involvement [23]. Nearly two out of three 12-14-year-old school children demonstrated some degree of DD on MRI in a cross-sectional study of 439 subjects [24]. A recent meta-analysis established a pooled prevalence of DD on MRI in 44% and 22% of adolescents with or without LBP, respectively [25].

Previous research has shown that degenerative changes of lumbar intervertebral discs are common in adolescents after the pubertal growth spurt. Little is known about the onset and development of these changes during growth, or their relevance to the clinical symptom of LBP. In the present study, our primary objectives were 1) to explore the SI changes in lumbar intervertebral discs on MRI during growth in a group of healthy school children, and 2) to investigate whether these changes were correlated to the clinical symptom of LBP.

## Methods

### Subject recruitment and study flow (Fig 1)

In 1994, we aimed to recruit a cohort of 100 school children for a longitudinal follow-up study on the natural history of lumbar intervertebral discs in healthy children. Six elementary schools from a total of 71 were randomly chosen from the urban capital area of Helsinki. All 2nd graders with an even birth date were invited to participate via a letter to their parents. Using this criterium, 208 out of 408 children were eligible; of them 108 were interested in participating. The baseline examination was performed at the age of 8–9 years (Y8) with follow-up examinations at the ages of 11–12 years (Y12) and 18–19 years (Y19). All time points included a semi-structured interview, a clinical examination, and an MRI investigation of the lumbar spine.

This study was conducted according to the Declaration of Helsinki for research on human participants. The ethical approval was granted by the Ethics Committee of the Invalid

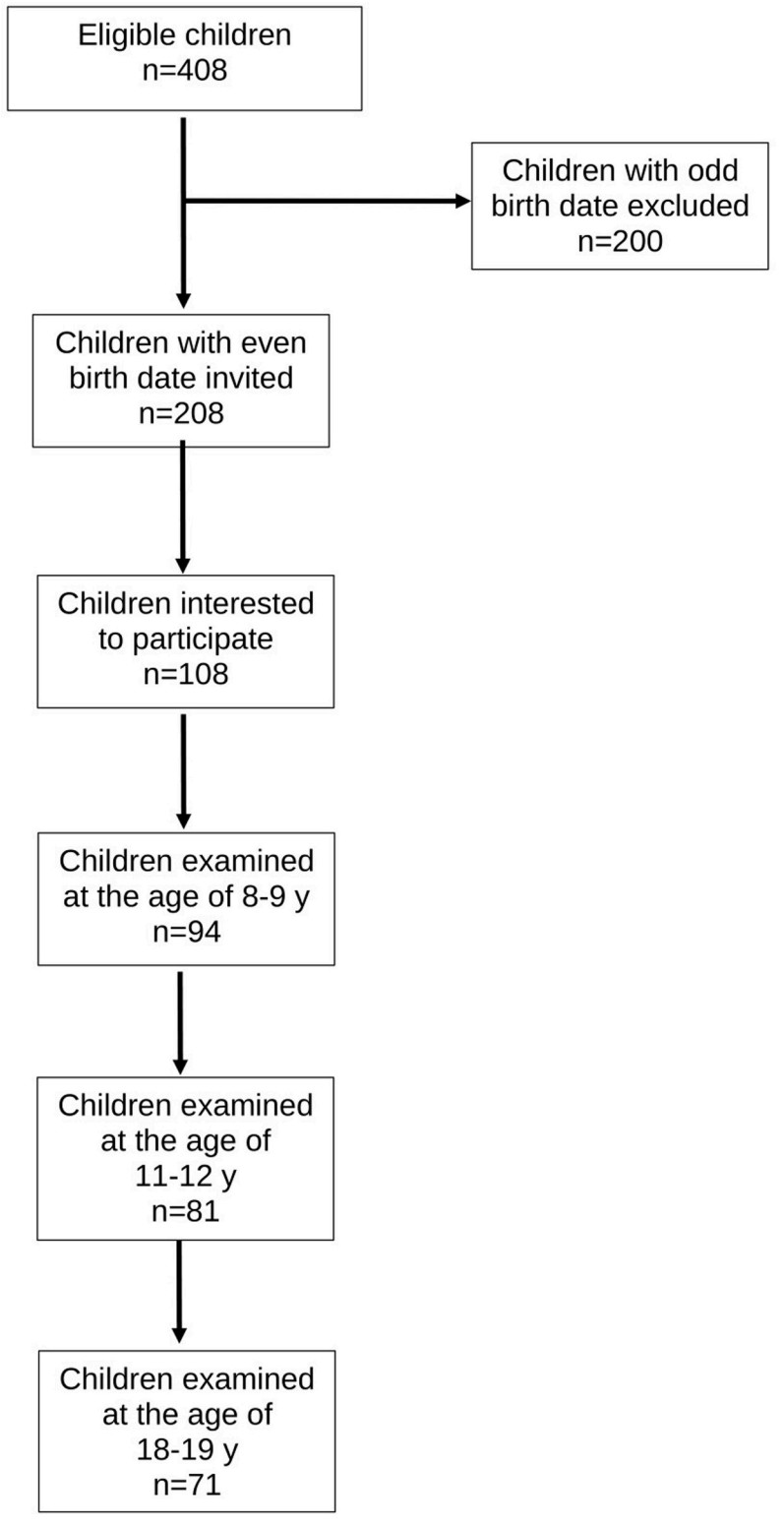

**Fig 1. Flow diagram of the study.**

Foundation on January 22, 1993; the study was registered to the Research Institute of the Invalid Foundation (later Research Institute Orton) on April 7, 1993. The study protocol was approved by the authorities of the Helsinki school district on August 30, 1993. A written informed consent from the parents of each child was obtained before commencement of the study.

## Semi-structured interview

In the semi-structured interview, the participants were asked whether they had ever had LBP without associated trauma. At Y19, the interview included more detailed information about the LBP (last week/last month/last year/earlier) and recorded a possible contact to a physician for the LBP symptom. To precisely localize the anatomical area of interest the participants were shown a body map highlighting the lumbar area. At Y19, the participants were asked whether they smoked, and if yes, how many cigarettes per day for how long.

## Clinical examination

The subject's height and weight were measured with a stadiometer and a balance-beam scale, respectively. Body Mass Index (BMI) for Y8 and Y12 was defined using the ISO-BMI formula taking into account the child´s age and gender; for Y19 we used the standard formula of BMI calculation for adults (weight in kilograms divided by the square of the height in meters). The clinical examination focused on identifying signs of scoliosis, functional leg length inequality, hamstring tightness, and reflex abnormalities.

## MRI investigation

At Y8 and Y12, the MRI investigation was obtained with a high-field 1.0T scanner (Siemens Magnetom, Siemens, Erlangen, Germany) using a dedicated spine coil. Only T2-weighted sagittal images were acquired with the following imaging parameters: TR 2500 ms, TE 80/15 ms, FOV 260, image matrix 256x256, slice thickness 4.0 mm, slice interval 4.4 mm, acq 1. At Y19, the MRI investigation was performed with a high-field 1.5T MRI scanner and a dedicated spine coil (Siemens Symphony, Siemens, Erlangen, Germany). The imaging parameters were as follows: TR 4630 ms, TE 107ms, FOV 280, image matrix 384x288, slice thickness 4.0 mm, slice interval 4.4 mm, acq.2.

The SI of the three lowest intervertebral discs (L3/L4, L4/L5 and L5/S1) was assessed both qualitatively and quantitatively from sagittal T2-weighted midline images. For the qualitative visual evaluation, a modified Schneiderman classification [26] was used. The discs were categorized as *Bright* with a preserved SI, *Speckled* with a heterogeneously decreased SI, or *Dark* with a diffuse loss of SI (Fig 2). A musculoskeletal radiologist (third author) and a spine surgeon (first author) independently graded the discs without any knowledge of the subject´s clinical characteristics. In case of discrepancy, the assessment of the third evaluator (second author) was used for consensus. This turned out to be necessary in 38 of the 732 discs (5.2%). The inter-rater agreement (Scott/Fleiss agreement coefficients with ordinal weights) for the three intervertebral levels combined at Y19 (with the highest prevalence of disc changes) ranged from 0.70 to 0.89 describing a substantial to almost perfect agreement [27].

The SI of the intervertebral disc was assessed quantitatively by a computerized method with a region of interest (ROI) marked digitally from each nucleus pulposus. The ROI in each individual disc was marked using a freehand technique according to the estimated area of the nucleus pulposus. As an internal reference the SI of the adjacent cerebrospinal fluid (CSF) was used for a disc to CSF -SI ratio [28]. For the ROI of the CSF at every level, the area in the anterior dural sac immediately posterior to the disc was chosen to exclude the effect of the nerve

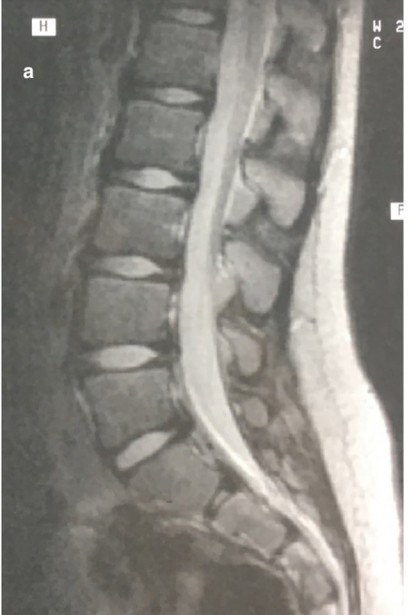
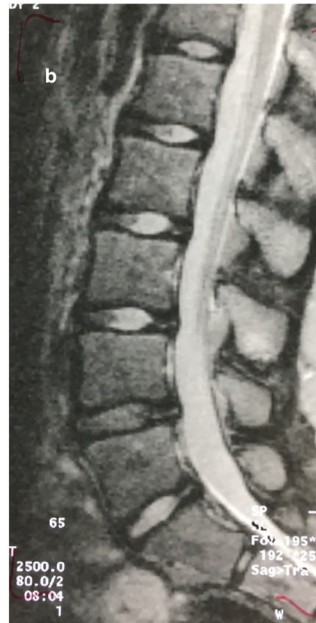
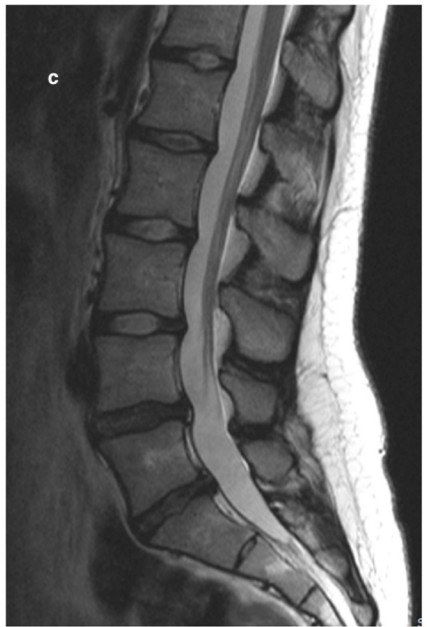

**Fig 2. The evolution of disc changes throughout the study period in one study participant.** The MRI images illustrate the visual assessment of the signal intensity (SI) of the intervertebral disc using the Schneiderman classification. At Y8, all lumbar discs presented with a *Bright* nucleus pulposus (a). At Y12, the L4/L5 disc was graded *Speckled* (b), and at Y19, both L4/L5 and L5/S1 discs were graded *Dark* (c).

roots. The measurements were performed by a musculoskeletal radiologist (third author) and a physician (sixth author) trained with the measurement technique.

## Data analysis

The descriptive statistics are presented as means with standard deviations (SD) or as counts with percentages. Repeated measures were analysed using generalising estimating equations (GEE) models with the unstructured correlation structure. Generalized estimating equations were developed as an extension of the general linear model (e.g., OLS regression analysis) to analyze longitudinal and other correlated data. GEE models take into account the correlation between repeated measurements in the same subject; models do not require complete data and can be fit even when individuals do not have observations at all time points. In case of violation of the assumptions (e.g., non-normality) for continuous variables, a bootstrap-type method or Monte Carlo p values (small number of observations) for categorical variables were used. The normality of variables was evaluated graphically and by using the Shapiro–Wilk W test. No adjustment was made for multiple testing. Stata 17.0 (StataCorp LP; College Station, Texas, USA) statistical package was used for the analysis.

## Results

Of the 108 children expressing interest in the study, 94 (46 females and 48 males) eventually participated. Two children did not want to go through the MRI investigation at Y8 but were included in the analysis with clinical data and MRI investigations at the two later time points. At Y12 and Y19, 81 and 71 study subjects participated in the interview, clinical examination and MRI resulting in 86% and 76% follow-up, respectively.

Table 1 gives a more detailed description of our study participants. The only statistically significant difference between sexes was seen at Y19 when males were significantly taller and

**Table 1. Description of study participants.**

|  | Y8 (N = 94) | Y12 (N = 81) | Y19 (N = 71) |
|---|---|---|---|
| **Mean age (SD), y** | 8.5(0.4) | 11.9 (0.5) | 19.3 (0.6) |
| **Female: Male, N** | 46:48 | 39:42 | 35:36 |
| **Mean height (SD), cm** |  |  |  |
| **Female** | 131 (6) | 152 (8) | 165 (7) |
| **Male** | 133 (7) | 152 (8) | 180* (7) |
| **Mean weight (SD), kg** |  |  |  |
| **Female** | 28.5 (6.0) | 42.0 (9.9) | 60.8 (12.0) |
| **Male** | 28.8 (5.5) | 43.0 (10.7) | 73.7* (17.2) |
| **Mean BMI (SD)** |  |  |  |
| **Female** | 16.4 (2.5) | 18.1 (3.3) | 22.3 (4.3) |
| **Male** | 16.1 (2.2) | 18.5 (3.4) | 22.8 (5.1) |
| **LBP, % (95% CI)** |  |  |  |
| **All**** | 6 (2 to 13) | 13 (7 to 23) | 54 (42 to 66) |
| **Female** | 9 (2 to 21) | 8 (2 to 20) | 56 (38 to 73) |
| **Male** | 4 (1 to 15) | 19 (9 to 34) | 53 (35 to 70) |

*p<0.001 according to sex

**p<0.001 for all participants according to age

weighted more than females (p<0.001). The growth for females between Y8 and Y12 was 21 cm (95% CI: 18 to 23; p<0.001) for a relative growth of 1.16 (95% CI: 1.13 to 1.18); and between Y12 and Y19 13 cm (95% CI: 10 to 17; p<0.001) for a relative growth of 1.09 (95% CI: 1.07–1.11). For males the growth between Y8 and Y12 was 19 cm (95% CI: 15 to 22, p<0.001) for a relative growth of 1.14 (95% CI: 1.11 to 1.16), and between Y12 and Y19 28 cm (95% CI: 25 to 31, p<0.001) for a relative growth of 1.18 (95% CI: 1.16 to 1.21). Between Y12 and Y19 the relative growth of males was significantly more than that of females (p<0.001).

## Occurrence of LBP

By the age of 19, 54% of the participants had experienced LBP without associated trauma. The increase in occurrence of LBP with age for the whole study population was statistically significant (p<0.001). Table 1 for the occurrence of LBP at different study time points.

## Visual assessment of disc changes and their correlation to LBP

In general, the L3/L4 disc remained stable throughout the study period; while some changes were noticed at the L4/L5 disc, most of the changes developed in the L5/S1 disc. Progression of at least one grade at L4/L5 and L5/S1 was demonstrated in 12% and 20% of participants, respectively, from Y8 to Y19. Specifically, at Y8, 21 of the 276 discs (7.6%) were graded as *Speckled* in 17 participants (18%); four participants had two-level involvement. Nine discs (3.7%) demonstrated *Speckled* pattern at Y12 in eight participants with two-level involvement in one participant. By Y19, 37 of the 213 discs (17.4%) demonstrated changes with 11 discs (5.2%) graded as *Dark*. Disc changes were noticed in 27 participants (38%) with two-level involvement in eight participants and one participant with a three-level involvement. Two participants had *Dark* discs both at L4/L5 and L5/S1 levels.

No association was found between visual assessment of the SI (*Bright—Speckled—Dark*) and LBP when the most degenerated disc at any level was selected for analysis (Table 2).

**Table 2. Association of the visual assessment of the most degenerated disc to self-reported LBP.**

| | No LBP N (%) | LBP N (%) | p-value |
|---|---|---|---|
| **Schneiderman score** | | | |
| **Y8** | | | 0.91 |
| *Bright* | 70 (81) | 5 (83) | |
| *Speckled* | 16 (19) | 1 (17) | |
| *Dark* | 0 | 0 | |
| **Y12** | | | 0.93 |
| *Bright* | 63 (90) | 10 (91) | |
| *Speckled* | 7 (10) | 1 (9) | |
| *Dark* | 0 | 0 | |
| **Y19** | | | 0.98 |
| *Bright* | 20 (61) | 24 (63) | |
| *Speckled* | 10 (30) | 8 (21) | |
| *Dark* | 3 (9) | 6 (16) | |

## Disc to cerebrospinal fluid -SI ratio and its association to LBP

In the disc to CSF -SI ratio, no significant changes were noticed between Y8 and Y12, whereas the ratio markedly decreased by Y19 (Fig 3). No statistically significant difference in the disc to CSF -SI ratio between participants with or without LBP was noticed at any level as illustrated in Fig 3.

At Y19, BMI showed a statistically significant association to the disc to CSF -SI ratio at the L5/S1 level (p = 0.035); no other statistically significant associations emerged at any of the disc levels at the different study time points (Table 3).

## Discussion

In the present study, we describe the natural history of lumbar intervertebral discs (L3/L4, L4/L5 and L5/S1) from childhood to early adulthood and examine the association of changes in the disc SI to the clinical symptom of LBP. At the age of 8–9 years, 18% of the participants presented with MRI findings that have traditionally been considered early signs of degeneration. At the age of 18–19 years, 17% of the discs demonstrated SI changes in 38% of the participants. By this time 54% of the participants had experienced LBP without associated trauma. The disc SI changes did not associate with the presence of LBP in childhood, adolescence or early adulthood.

The strength of the present study lies in its design; to our knowledge, this is the first longitudinal study assessing the natural history of lumbar intervertebral discs with MRI from childhood to early adulthood. However, several limitations need to be considered when interpreting our results.

Healthy school children with an even birth date were recruited to this study. The rationale was to have children with an odd birth date as possible controls with no history of repeated inquiries about LBP. Of the 208 eligible participants, 108 expressed interest in the study. It is entirely possible that a family history of LBP affected their decision to participate. In a population-based twin study, Hestbaek et al. showed a significant genetic influence on the development of early LBP [29]. While some research has shown significant association between parental and child back pain [30, 31] with children potentially modeling the symptoms of their parents [30], others have found no evidence of learned pain behavior in children [32]. Based on a recent systematic review, up to 47% of children and adolescents report either occasional

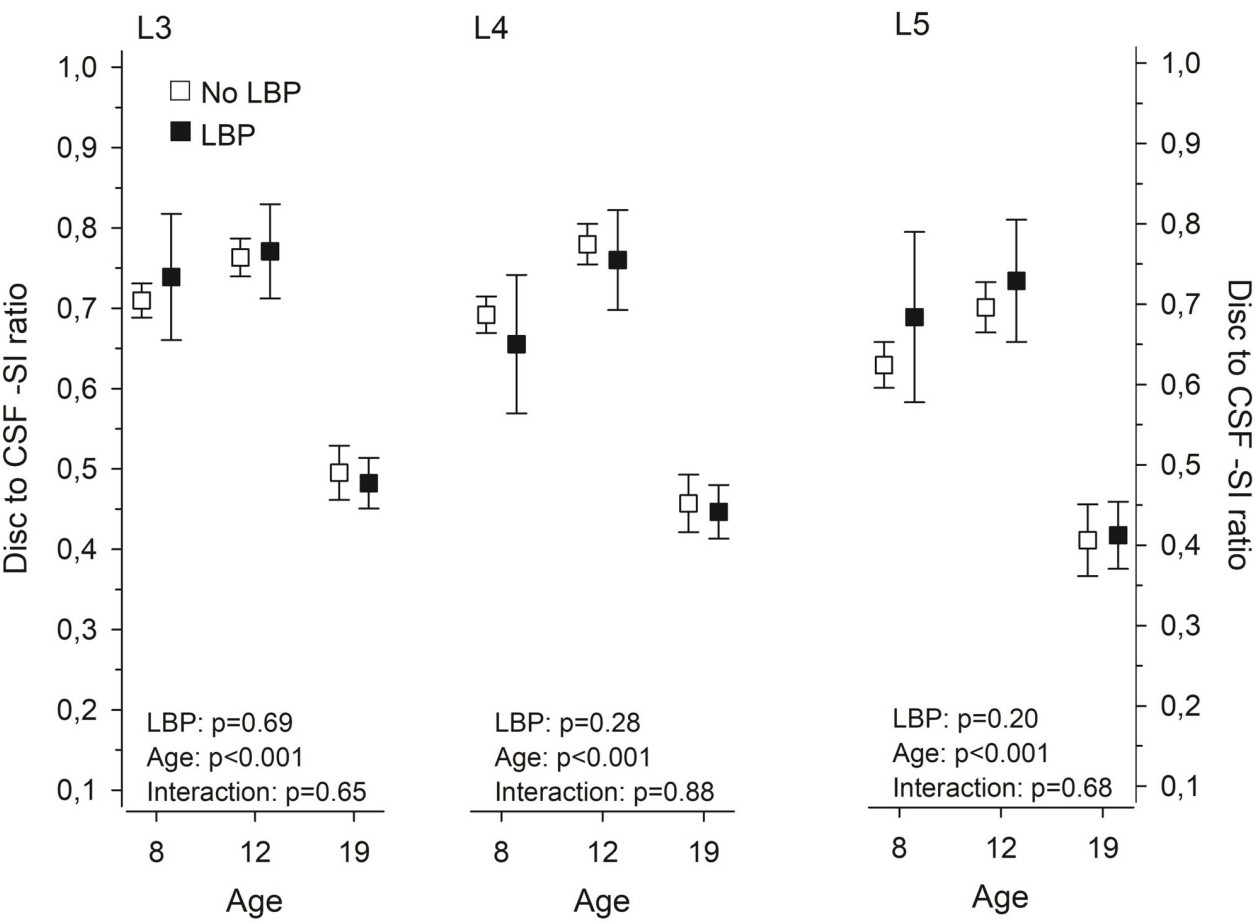

**Fig 3. Computerized measurement of the disc signal intensity (SI) to cerebrospinal fluid SI–ratio at 8, 12, and 19 years in subjects with or without LBP.** Means with whiskers representing 95 per cent confidence intervals.

or recurrent LBP [2]. The prevalence estimates surpass 50% at the age of 18 to 20 years [5]. The prevalence of LBP in the present study, 54% by the age of 18–19 years, corresponds with previous studies on larger cohorts reporting lifetime prevalence from 40% to 79% by adolescence and early adulthood [32–37].

One could speculate that repeated inquiries about the presence of LBP will result in increased reporting. We tried to avoid this by keeping the discussion of LBP limited. The participants and their parents were informed that the main purpose of the study was to investigate the growth of the lumbar spine in children ("Healthy spine in a growing child"). Moreover,

**Table 3. The Spearman correlation between BMI and Disc to CSF -SI ratio.**

|  | Y8 | Y12 | Y19 |
|---|---|---|---|
|  | (N = 92) | (N = 81) | (N = 71) |
|  | r (95% CI) | r (95% CI) | r (95% CI) |
| **L3/L4** | 0.16 (-0.05 to 0.28) | -0.03 (-0.25 to 0.19) | 0.05 (-0.19 to 0.28) |
| **L4/L5** | 0.00 (-0.21 to 0.21) | -0.02 (-0.24 to 0.20) | -0.04 (-0.27 to 0.20) |
| **L5/S1** | -0.14 (-0.33 to 0.07) | -0.06 (-0.27 to 0.17) | -0.25* (-0.46 to -0.02) |

* p = 0.035

self-reported LBP should be regarded with a degree of uncertainty. When studying children and adolescents, particularly about lifetime prevalence of LBP, memory decay may influence the results. Although high level of forgetfulness of previous LBP has been shown in children [3], this appears less significant for recurrent or more severe episodes [38]. It may well be that our participants had forgotten about previous less severe episodes, as at Y19 84% of participants with LBP reported pain either last week or last month. The pain intensity was not formally evaluated, but only seven participants (18% of those reporting LBP) had visited a physician suggesting moderate LBP with limited effect on the activities of daily life.

Ninety-four (94) children comprised our initial study group. At Y12, 13 participants were lost to follow-up. Four of them had mild MRI changes at Y8, and one had reported LBP without disc changes. At Y19, 23 participants were lost to follow-up. Only four of them had disc changes in the first and/or second MRI investigation; one of them had reported LBP at Y12. Three additional participants lost to follow-up had reported LBP at Y12 without disc changes. Most of the participants lost to follow-up at Y19 had not experienced LBP or demonstrated disc changes at the previous examinations.

Our study covered the evolution of MRI technology; the first two MRI investigations were performed with a 1.0T scanner and the last one with a 1.5T scanner. Some evidence suggests that the field strength of the MRI equipment does not have a significant effect on the assessment of spinal morphology [39]. Our main interest was the SI of the intervertebral disc at the three lowest lumbar levels. While SI can be assessed qualitatively and quantitatively, it is dependent on a variety of technical and patient-related factors, e.g., the distance of the object of interest (in our case the intervertebral disc) from the surface coil. As the absolute contrast and SI on each MRI slice is determined by the brightest pixel, non-standardized SI measurements would introduce a significant error when comparing different study participants and time points. To minimize the effect of confounding factors and to compare the SI within and between participants at different time points, we used the SI of the adjacent CSF as a reference for a relative SI. The adjacent CSF has the advantage of being close to the intervertebral disc and having a relatively constant SI [40, 41]. The disc to CSF -SI ratio has proven sensitive to early disc changes in young subjects [28]. All MRI investigations were performed in the morning to prevent possible diurnal variation of the SI.

No grading system for early intervertebral disc changes in children and adolescents has been introduced. We used the Schneiderman classification [26], albeit slightly modified as we did not expect advanced disc changes in our young study participants. The computerized disc to CSF -SI ratio remained relatively stable between Y8 and Y12 with a marked decrease at Y19 reflecting the results of the visual assessment.

At the age of 8–9 years, 18% of our participants presented with mild disc changes (*Speckled*) on T2-weighted MRI-images. By the age of 18–19 years 38% of the participants had disc changes (*Speckled* or *Dark*). A recent systematic review and meta-analysis of abnormalities in the pediatric spine on MRI demonstrated a 22% (95%CI 9%-38%) pooled prevalence of DD in children without LBP and 44% (95%CI 23%-65%) in children with LBP [25]. In a prospective cross-sectional study of 439 schoolchildren, Kjaer et al found DD on MRI in approximately one third of their 12-14-year-old participants [24]. Our results are in line with these previous studies.

In the present study, we did not find an association between disc SI changes and self-reported LBP. This is contrary to a systematic review reporting higher prevalence of DD in children with LBP [25]. It is noteworthy that the studies included in the meta-analysis were performed in a hospital setting, possibly implying a more severe symptom state and overestimation of the difference between children with and without LBP. The earliest MRI studies from the 1990s suggested that young subjects with early degenerative changes are more prone

to LBP in the future [21, 22, 42]. In a cross-sectional MRI study on young adults, moderately degenerated discs were likely to be associated with more severe clinical symptoms compared to mildly degenerated discs [18]. However, DD was also found in one third of asymptomatic subjects. In the present study, no association between the most degenerated disc (defined by the Schneiderman classification) regardless of disc level and self-reported LBP was found. In a cross-sectional study of 439 12-14-year-old adolescents, most disc-related findings were only weakly associated with LBP; statistically significant associations for boys emerged in the upper and for girls in the lower lumbar levels [24]. In our study, only the three lowest lumbar levels were analyzed as significant disc changes in the upper lumbar spine in this group of young participants were unlikely.

Only T2-weighted sagittal images were obtained to reduce the scanning time in our young participants. Thus, our analysis was limited to changes in the SI of the intervertebral discs, which might have caused us to overlook other morphological changes related to LBP. For example, a more significant association between disc bulge and LBP has been suggested in younger adults compared to older subjects [19]. DD, however, is the most common structural abnormality in the pediatric spine on MRI [25]. and thus of special interest clinically.

For a clinically meaningful analysis of the evolution of disc height we would have needed a standing lumbar spine x-ray at each of the study time points. This would have exposed our participants to unnecessary radiation and was not included in the study design. Moreover, the validity of absolute disc height, alone or in combination with other measures, as an indicator of early DD has been questioned [43]. Pfirrmann et al., in their MRI investigation of 70 asymptomatic adults, concluded that the association of DD and disc height was stronger in older individuals compared to younger subjects [44].

For the disc to CSF -SI ratio, higher BMI at Y19 correlated statistically significantly with lower relative SI at the L5/S1 level, although the correlation was only fair. High BMI at 16 years of age has previously been shown to be associated with lumbar DD among young males [45]. In another population-based cross-sectional study, overweight or obese adolescents and young adults had more severe DD than underweight or normal-weight individuals [23]. Compared to these previous findings, our results are more in line with those of van den Heuvel et al. who found no association between increased BMI and disc SI in their 9-year-old subjects [46].

We did not perform a formal power analysis to define the number of participants. Due to financial constraints we had to restrict our study population to approximately 100 children and 300 MRI investigations; 94 children formed the initial study group, 13 and 23 participants were lost to follow-up at Y12 and Y19, respectively. Thus, it is entirely possible that due to a small number of participants our study did not have enough power to detect possible associations between disc SI changes and LBP.

## Conclusions

Our study adds to the existing evidence in providing data on the natural history of intervertebral disc morphology in a group of healthy subjects from childhood to early adulthood. The prevalence of LBP increased significantly with age reaching 54% by the age of 18–19 years. Some mild disc SI changes on MRI were seen at the age of 8–9 years in 18% of our participants; after the growth spurt 38% of our 18-19-year-old participants demonstrated disc SI changes. In this small study population, these disc SI changes did not have an association with LBP. If adolescents and young adults complain of LBP without symptoms and signs of specific etiology, it is unlikely that an MRI investigation will benefit the diagnostic workup or therapy, and it may eventually lead to poorer health outcomes due to unfounded conviction that incidental and innocuous findings on MRI are the cause of pain.

## Supporting information

**S1 File. Supporting data set.**
(XLSX)

## Author Contributions

**Conceptualization:** Dietrich Schlenzka, Kalevi Österman.

**Data curation:** Leena Ristolainen, Hannu Kautiainen.

**Formal analysis:** Teija Lund, Dietrich Schlenzka, Martina Lohman, Hannu Kautiainen.

**Funding acquisition:** Dietrich Schlenzka, Kalevi Österman.

**Investigation:** Teija Lund, Dietrich Schlenzka, Martina Lohman, Erkko Klemetti.

**Methodology:** Teija Lund, Dietrich Schlenzka, Martina Lohman, Hannu Kautiainen, Kalevi Österman.

**Project administration:** Leena Ristolainen.

**Resources:** Dietrich Schlenzka, Leena Ristolainen.

**Supervision:** Dietrich Schlenzka.

**Visualization:** Hannu Kautiainen.

**Writing – original draft:** Teija Lund.

**Writing – review & editing:** Teija Lund, Dietrich Schlenzka, Martina Lohman, Leena Ristolainen, Hannu Kautiainen, Erkko Klemetti, Kalevi Österman.

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
