## [Decision Letter · Decision Letter 0]

17 Mar 2022

PONE-D-22-02909The intervertebral disc during growth: changes on magnetic resonance imaging and their relevance to low back painPLOS ONE

Dear Dr. Lund,

Thank you for submitting your manuscript to PLOS ONE. After careful consideration, we feel that it has merit but does not fully meet PLOS ONE’s publication criteria as it currently stands. Therefore, we invite you to submit a revised version of the manuscript that addresses the points raised during the review process.

The reviewer panel is of the impression that the study does contribute to the field, especially by providing new data on natural progression of the intervertebral disc conditions in a young cohort. However, there are a few important statistical and methodological issues that need to be addressed in detail before the manuscript can be accepted for publication. Namely, a proper power of study, a detailed statistical evaluation of low back pain prevalence, consistency in MRI data, and the inclusion of clinical and morphological spinal parameters. Please refer to the reviewer comments for more details on these questions.

We look forward to receiving your revised manuscript.

Kind regards,

Alejandro A. Espinoza Orías, PhD

Academic Editor

PLOS ONE

Journal Requirements:

3. We noted in your submission details that a portion of your manuscript may have been presented or published elsewhere. [Our long-term follow-up (currently under review for publication) describes a subset (approx 50%) of the original study group. The analysis approaches the data from a different perspective; no text or figures in the present manuscript have been taken from the manuscript on the long-term results.] Please clarify whether this publication was peer-reviewed and formally published. If this work was previously peer-reviewed and published, in the cover letter please provide the reason that this work does not constitute dual publication and should be included in the current manuscript.

Reviewers' comments:

Reviewer's Responses to Questions

**Comments to the Author**

1. Is the manuscript technically sound, and do the data support the conclusions?

Reviewer #1: Partly

Reviewer #2: Yes

2. Has the statistical analysis been performed appropriately and rigorously? 

Reviewer #1: No

Reviewer #2: Yes

3. Have the authors made all data underlying the findings in their manuscript fully available?

Reviewer #1: No

Reviewer #2: Yes

4. Is the manuscript presented in an intelligible fashion and written in standard English?

Reviewer #1: Yes

Reviewer #2: Yes

5. Review Comments to the Author

Reviewer #1: “In this longitudinal cohort study [of at least 71 children (8-19 yr)], the objective was to determine the natural history of disc changes from childhood to early adulthood, and the possible association of these changes to LBP.” Outcomes included a survey for life-long back pain and an MRI-based classification of the disc and csf. Prevalence of LBP increased with aging but was not associated to disc hydration intensity. Some key structural measures that are possible were not determined and some key statistical metrics are either not included or explicitly stated.

Conceptual comments

None.

Major technical comments:

• Two different MRIs were used in the study, where the first two scans were of Y9 and T12 and the second scanner was used for Y19. The assumption is made that the relative change between disc and CSF would account for machine/parameter/calibration differences but this assumes a linear relationship and may not be the case. Disc/csf ratios of Y9 or Y12 on the latter scanner (1.5T) must be corroborated to those in the former (1.0T)? Otherwise, the age-related effect is due to use of a different scanner and parameter differences.

• References >43 are missing.

• “The most significant increase occurred after Y12 with 54% of participants having experienced LBP. . .” but the prevalence of LBP was not compared statistically. A Chi-square or something similar is necessary to draw this conclusion with respect to age or degree intensity and is necessary for all of the nominal data in the manuscript.

• What is the disc height, disc concavity and/or spinal curvature (Cobb angle)? Are these correlated to LBP? Do they change with age?

• Does disc morphology, intensity changes or LBP with aging remain following normalization to BMI or smoking?

• The sample size for the following statement appears unclear: “The only significant finding was that all participants with a dark disc at L4/L5 level at Y19 (n=4) reported LBP (p=0.048).” In Table 2, there were n=6 participants with LBP. Secondly, what was the control group to determine this finding: Y8 or Y12 of the same participants, incidence of LBP in dark disc at Y19?

• How did incidence of LBP in an individual relate to disc morphology and hydration?

• “The prevalence of LBP increased significantly after the pubertal growth spurt reaching 54% by the age of 18-19 years.” This statement is not supported by any particular data and requires a comparison showing that the height change from 12-19 was greater than 12-9 in this cohort. Y19 smoked while all others did not. Any number of other reasons may have engendered LBP.

• Include LBP incidence per child.

Minor comments

• “At the age of 8-9 years, 18% of the participants presented with MRI findings that have traditionally been considered early signs of degeneration.” What outcome is this and from where in Table 2 is this data? A more specific statement is needed that explains that the “traditional” metric is disc intensity distribution for all children irrespective of the LBP incidence.

• What was the level of activity of the participants, e.g., sports, etc.? Many people were confined indoors because of COVID19. When were these latter data collected?

Reviewer #2: The manuscript details a very interesting study of children over a decade of life attempting to investigate relationships between low back pain and structural changes within the disc observed on MRI. The longitudinal nature of the investigation is impressive, and this data will certainly be a great contribution to the field as studies of this nature are lacking. Please see below for specific comments and suggestions:

1. Abstract – claiming that disc changes on MRI “are not associated with the presence of LBP” is somewhat misleading – the authors state that there was a significant finding that participants with a “dark” disc at L45 at Y19 reported back pain.

2. Methods - For the signal intensity characterization on MRI – what were the dimensions of the ROIs used? Or were they scaled to the size of the disc?

3. Table 1 – were there statistically significant differences in demographics at any age?

4. Were the children all from families with similar socioeconomic status? Studies have suggested socioeconomic status may be a factor in the risk for development of back pain.

5. Results – Prevalence of Back Pain – Was smoking associated with an increased prevalence of back pain? Was back pain more prevalent in males versus females at any age group? Given this large and unique data set, these points I think would be of interest to readers.

6. MRI Results – it seems data from only L3-S1 was utilized. Is there a reason the L1-2 and L2-L3 discs were excluded from analysis?

7. Results, Figure 2 – labels need to be added to this Figure, are these scans from the same patient at each time point? There does not appear to be a figure legend included.

6. PLOS authors have the option to publish the peer review history of their article (what does this mean?). If published, this will include your full peer review and any attached files.

Reviewer #1: No

Reviewer #2: No

---

## [Author Response · Author response to Decision Letter 0]

6 Apr 2022

We appreciate the opportunity to re-submit our revised manuscript to PLOS ONE and are grateful for the thorough and insightful comments of your reviewers. We have addressed their comments in this response letter and made changes in the manuscript accordingly. We believe these changes have made our manuscript and our message stronger and clearer.

Editor´s comments:

We have addressed the reviewers’ comments regarding a statistical evaluation of LBP prevalence, consistency in MRI data, and the inclusion of additional clinical and morphological spinal parameters in our response below. 

We did not perform a formal power analysis. When we started the study, MRI investigations were still relatively rare and significantly more expensive than nowadays. Our funding allowed us to perform three consecutive MRIs on around 100 participants (300 MRIs) which defined the size of the original study group. Although the power of our study may not be enough for any definitive conclusions, we believe the study is still unique in its design and scope.

1. We have re-checked that our manuscript meets the PLOS ONE requirements.

2. We have revised the funding information. The grants received do not have an official grant number; they are listed using a specific project number (A2500/465 for the current research project).

3. The long-term follow-up consisting of a sub-population of the original study population (reported here) has been accepted for publication in the European Spine Journal (peer-reviewed and published online before print). We have included the online version of the manuscript to this response letter as a supplemental file for your review.

4. We have included a supplemental file with a data set of the results described in the present manuscript. We continue to analyze the data and intend to report additional results in the future.

Reviewer´s comments:

Reviewer #1:

1. This is a valid concern and a possible source of error. Our study spanned the evolution of MRI technology with the first two MRI studies performed with a 1.0T scanner and the last one with a 1.5T scanner. The contrast on MRI depends on the field strength and imaging parameters; all our MRI studies were performed using a high-field scanner. Using different scanners introduces a possibility of error but standardizing the SI measurements with a specific internal reference (the adjacent CSF) in each study subject for all disc levels separately minimizes the risk. The same method has been used in the few longitudinal studies published to date using MRI scanners with different field strengths.

The brightest pixel determines the contrast and SI on each MRI slice. Thus, standardization to a tissue or liquid (such as CSF) that is homogeneous e.g., biologically and regarding the relaxation time, is mandatory. Our reference was measured separately for all disc levels on each study participant to ensure maximum accuracy. We acknowledge that non-standardized SI measurements would have introduced a significant error when comparing different study participants and time points.

In the revised manuscript, we have described our measurement technique for the relative signal intensity in more detail; moreover, we have strengthened the discussion on this topic when discussing the limitations of our study.

2. We thank the reviewer for pointing out an unfortunate mix-up in our reference list. We have addressed this, and the reference list is now correct.

3. This is an important remark by the reviewer. We have added information on the prevalence of LBP at each study time point to Table 1. 

4. The decrease in disc SI on MRI is the most common finding in children and adolescents with increasing age. We decided to concentrate on this parameter in our primary analysis. No universally accepted grading system for early intervertebral disc changes in children and adolescents has been established. Our initial study design was based on the Schneiderman classification introduced in 1987; the Schneiderman classification focuses on the disc SI. We are currently in the process of re-assessing the MRIs using the Pfirrmann classification (introduced in 2001) that considers not only the disc SI but also the disc height. However, we did not expect major disc height reduction in our young study population. We did not evaluate disc concavity but recognize the wide variability of disc shapes seen in real life; however, no classification has addressed this phenomenon making its analysis difficult. The effect of spinal curvature (lumbar lordosis) on disc degeneration is an interesting question. We only had supine MRI images and thus could not evaluate the development of lumbar lordosis during growth. For a reliable and meaningful measure of lumbar lordosis we would have needed standing x-rays from our study participants in each study time point. When planning the study, we did not expect the ethical committee to accept this, and thus did not include it in our study protocol.

5. In the present manuscript, we decided to concentrate on the disc SI and its association to LBP; this decision was made consciously to keep the reporting concise and the message clear. We hope that this decision is acceptable. We have information on the BMI and smoking and are currently in the process of analyzing the data in relation to disc SI and LBP. We hope to report our findings in further manuscripts. In the present manuscript, we have deleted the information on smoking from Table 1.

6. We have revised Table 2 such that the message comes across clearer. The revised Results section also has a more detailed description of the disc findings at the three study time points. At ages 8 and 12, no dark discs were found in any of the study subjects. At age 19, dark disc at L4/L5 was seen in 4 participants and at L5/S1 in 7 participants. Two participants had dark discs both at L4/L5 and L5/S1 levels leaving 9 participants with dark discs at age 19. The only significant finding was that the 4 participants with a dark disc at L4/L5 level reported LBP. 

7. We have added the results of our analysis of the relative change in height to the results section. The results confirm that by age 12 girls had already entered the growth spurt while with boys more growth occurred after the age of 12 years. We did not have a control group as the study design was a longitudinal observational study. We agree with the reviewer that there are many factors that might affect both the changes in disc SI and the occurrence of LBP. We are in the process of analyzing the effect of e.g., BMI, smoking, physical activity level, family socio-economic status, family history of LBP, and hope to report our results in the future.

8. We have included the prevalence of LBP in Table 1. The supplemental file on the data set reported herein gives detailed information about LBP at an individual level. 

9. A more detailed description has been added to the Results section.

10. We will analyze the effect of activity level both to disc SI and LBP. We started our study in 1994-1995 so the final study time point was before the COVID era.

Reviewer #2:

1. We thank the reviewer for this comment. It is true that this finding was borderline significant. We did not want to put too much emphasis on it as it was only 4 participants, but in this revised manuscript, we have added this information to the abstract and the body of the text as well.

2. The dimensions of the ROIs for the disc SI measurement were scaled according to the size of the nucleus pulposus of each disc. We have added a more detailed description of the measurement method to the revised manuscript.

3. We have added (Table 1) a statistical analysis of growth between the different time points in girls and boys separately. Otherwise, the data on Table 1 is related to growth (a biological phenomenon) and as such are not suited for statistical analysis. Moreover, the Results section now includes a more detailed description of the relative growth indicating that by Y12 girls were already in the growth spurt phase while for boys this started later.

4. Our national education system is based on public schools, and thus majority of children from all socioeconomic classes attend the public school system. All six schools involved in the present study were part of the national school system. We have detailed information on the parental education and family income levels and intend to analyze this data in relation to LBP.

5. We agree with the reviewer that the data set is unique and hope to publish our results on the effect of gender, smoking, BMI, sports activities, family history of LBP etc. on both the disc SI and the occurrence of LBP. In the present manuscript, we decided to concentrate on the development of disc SI from childhood to early adulthood with special emphasis on self-reported LBP. We hope that this decision is acceptable.

6. In the original study protocol, we decided to concentrate on the three lowest disc levels, as “degeneration” usually starts at the lowest lumbar levels, and we did not expect any major changes at the upper lumbar levels in our young study population. In the ongoing analysis, using the more recent Pfirrmann classification we have analyzed the SI of all the lumbar discs. The preliminary analysis corroborates our original hypothesis of disc changes occurring in the lowest lumbar levels up to early adulthood. 

7. We have delineated the legend for Figure 2 more clearly compared to the original manuscript. The labels have been added accordingly.

---

## [Decision Letter · Decision Letter 1]

20 Jun 2022

PONE-D-22-02909R1The intervertebral disc during growth: signal intensity changes on magnetic resonance imaging and their relevance to low back painPLOS ONE

Dear Dr. Lund,

Thank you for submitting your manuscript to PLOS ONE. After careful consideration, we feel that it has merit but does not fully meet PLOS ONE’s publication criteria as it currently stands. Therefore, we invite you to submit a revised version of the manuscript that addresses the points raised during the review process. The  reviewer panel was of the opinion that the statistics were not treated properly in the manuscript and that only minor superficial corrections were made to the initial submission. For example, the cohort of only four subjects with a dark disc needs more scrutiny and statistical support since it is a very small sample size. Please refer to the detailed comments by the reviewers for these and other questions. Adding a statistician to the author team to address these data treatment/analysis issues would be beneficial in this case.

We look forward to receiving your revised manuscript.

Kind regards,

Alejandro A. Espinoza Orías, PhD

Academic Editor

PLOS ONE

Reviewers' comments:

Reviewer's Responses to Questions

**Comments to the Author**

1. If the authors have adequately addressed your comments raised in a previous round of review and you feel that this manuscript is now acceptable for publication, you may indicate that here to bypass the “Comments to the Author” section, enter your conflict of interest statement in the “Confidential to Editor” section, and submit your "Accept" recommendation.

Reviewer #1: (No Response)

Reviewer #2: All comments have been addressed

Reviewer #3: (No Response)

2. Is the manuscript technically sound, and do the data support the conclusions?

Reviewer #1: No

Reviewer #2: Yes

Reviewer #3: (No Response)

3. Has the statistical analysis been performed appropriately and rigorously? 

Reviewer #1: No

Reviewer #2: Yes

Reviewer #3: (No Response)

4. Have the authors made all data underlying the findings in their manuscript fully available?

Reviewer #1: Yes

Reviewer #2: Yes

Reviewer #3: (No Response)

5. Is the manuscript presented in an intelligible fashion and written in standard English?

Reviewer #1: Yes

Reviewer #2: Yes

Reviewer #3: (No Response)

6. Review Comments to the Author

Reviewer #1: Overall, the authors have not addressed the comments or concerns of Reviewer 1.

There are no statistical comparisons listed in the tables.

A chi-square can determine whether age influenced the incidence of LBP (for nominal data) and this was not included.

The authors did not calculate disc height and therefore, could not respond to the subsequent questions.

The authors chose to ignore the potential influence of BMI on SI even after the reviewer asked for clarification.

“The only significant finding was that the 4 participants with a dark disc at L4/L5 level reported LBP.” This statement is not clearer because the comparison is not stated. There are three ages and three types of intensity. What is compared to what to draw this conclusion?

From Response: “We have added the results of our analysis of the relative change in height to the results section. The results confirm that by age 12 girls had already entered the growth spurt while with boys more growth occurred after the age of 12 years.” From manuscript: lines 203-209 show that the relative growth between males and females was similar and they were not compared statistically.

Separately, how can a growth of 19 cm have a p>0.001? It is likely less than 0.001.

Reviewer #2: (No Response)

Reviewer #3: Authors conduct a longitudinal cohort study to determine the history of disc changes from age 8 to 19 years old and evaluate the association between these changes and self-reported low back pain (LBP). They analyzed 208 participants and observe 54% prevalence of LBP at year 19. Also they observed the association between dark disc at L4/L5 level at year 19 and LBP.

1. Abstract/line 224. “all four participants with a dark disc at…” only four participants in this analysis? The sample size is too small. Please comment on the generalizability of this result. Also, what statistical test was performed here for sample size of 4? Is it suitable for such a small sample?

2. Line 332. Please clearly report the power analysis results so that readers know how to evaluate these insignificant results.

7. PLOS authors have the option to publish the peer review history of their article (what does this mean?). If published, this will include your full peer review and any attached files.

Reviewer #1: No

Reviewer #2: No

Reviewer #3: No

---

## [Author Response · Author response to Decision Letter 1]

31 Aug 2022

Re: PONE-D-22-02909R1 (The intervertebral disc during growth: signal intensity changes on magnetic resonance imaging and their relevance to low back pain)

Dear Dr. Espinoza Orias

We would like to take this opportunity to thank you and your reviewer´s for your continued commitment to make our manuscript better. Please see below our detailed responses to your and your reviewer´s comments.

Author response to the editor

Editor´s comment:

Adding a statistician to the author team to address these data treatment/analysis issues would be beneficial in this case.

Author response:

We have had an experienced professional medical statistician in our author team from the beginning (Dr Hannu Kautiainen). PubMed gives him a list of more than 850 peer-reviewed publications as a co-author. It would have been impossible to analyze and handle the vast amount of complex data without him.

Author response to reviewers

Reviewer 1:

Reviewer comment:

Overall, the authors have not addressed the comments or concerns of Reviewer 1.

Author response: 

We regret that our revisions to the abovementioned manuscript did not meet the standards of Reviewer 1. We have now carefully gone through the comments and responded to them to the best of our ability. We hope that this re-revision will address the reviewer´s concerns regarding our manuscript. Our initial plan for publication of the results was to concentrate on the association between changes in disc signal intensity and LBP first and then to elaborate on the association of e.g., demographics with disc signal intensity and LBP. We have now reconsidered our previous decision and included additional data in this re-revision of our manuscript.

Author action: 

Please see below our detailed responses and the actions we have taken based on the comments of Reviewer 1. 

Reviewer comment: 

There are no statistical comparisons listed in the tables.

Author response: 

The information in Table 1 describing the growth of the participants is not susceptible to statistical analysis. We have analyzed our baseline demographic data at each study time point separately for females and males; the results of this analysis have been included in the manuscript and in Table 1. Statistical analysis of the data in Table 2, please see below.

Author action: 

Table 1

Both in the manuscript and in Table 1, we have added information on the statistically significant differences between males and females throughout the study period. 

 Results (2nd chapter) in the re-revised manuscript state the following: The only statistically significant difference between sexes was seen at Y19 when males were significantly taller and weighted more than females (p<0.001). 

 In Table 1, the statistically significant findings between females and males have been highlighted with an asterisk (*).

 Regarding the statistical analysis of the occurrence of low back pain with age (Table 1), please see the separate comment below.

Table 2 

To avoid the problems of multiple testing, we analyzed the data presented in Table 2 according to the most degenerated disc regardless of the intervertebral level. With this analysis the previous statistically borderline significant (p=0.048) association between a dark disc at the L4/L5 level and LBP in four study subjects disappeared. We have revised the manuscript accordingly:

 We have deleted all information on the above-mentioned finding from the abstract and the text proper. Specifically, in the abstract, the conclusion now states: Changes in disc SI were not associated with the presence of LBP in childhood, adolescence or early adulthood. In the Results, we deleted the following statement: “When the disc levels were considered separately, the only significant finding was that all participants with a Dark disc at L4/L5 level at Y19 (n=4) reported LBP (p=0.048).” In the Discussion, the following statement was deleted: “The only statistically significant association between disc degeneration and LBP emerged in the four participants with a dark disc at L4/L5 level at the age of 19 years who all reported LBP.” In the revised manuscript, we stated “The most significant increase in both LBP and disc changes occurred after the pubertal growth spurt. The only statistically significant association between disc degeneration and LBP emerged in the four participants with a dark disc at L4/L5 level at the age of 19 years who all reported LBP.” This statement has now been modified as follows (Discussion, 1st chapter): The disc SI changes did not associate with the presence of LBP in childhood, adolescence or early adulthood. In the Conclusions, we deleted the following statement: “The only significant relationship between LBP and disc SI changes was noticed in the four participants with a dark disc at L4/L5 level at the age of 19 years.” The Conclusions now state: In this small study population, these disc SI changes did not have an association with LBP. 

 Based on the abovementioned analysis, we have significantly simplified Table 2; the p values are included in the Table.

Table 2. Association of the visual assessment of the most degenerated disc to self-reported LBP

 No LBP

N (%) LBP

N (%) p-value

Schneiderman score 

Y8 0.91

Bright 70 (81) 5 (83) 

Speckled 16 (19) 1 (17) 

Dark 0 0 

Y12 0.93

Bright 63 (90) 10 (91) 

Speckled 7 (10) 1 (9) 

Dark 0 0 

Y19 0.98

Bright 20 (61) 24 (63) 

Speckled 10 (30) 8 (21) 

Dark 3 (9) 6 (16) 

Reviewer comment: 

A chi-square can determine whether age influenced the incidence of LBP (for nominal data) and this was not included.

Author response: In the previous version of the manuscript, we only reported the confidence intervals.

Author action: 

We have now included the statistical analysis of the influence of age on the incidence of LBP both in the manuscript and in Table 1. The analysis was performed using the Monte Carlo p values for the whole study group, i.e., females and males together. The increased occurrence of LBP with age was found to be statistically significant.

 In the manuscript, the information is given as follows (Results, chapter “Occurrence of LBP): By the age of 19, 54% of the participants had experienced LBP without associated trauma. The increase in occurrence of LBP with age for the whole study population was statistically significant (p<0.001). Table 1 for the occurrence of LBP at different study time points.

 In Table 1, this statistical significance is illustrated by **.

Reviewer comment:

The authors did not calculate disc height and therefore, could not respond to the subsequent questions. 

Author response:

In the first revision round, Reviewer 1 addressed this issue as follows: Are these correlated to LBP? Do they change with age? The association of disc height to low back pain is an interesting line of research but was not our primary focus when we designed our study. Clinically meaningful evaluation of disc height from supine MRI images is limited; to achieve this goal we would have needed standing x-rays from our study participants at each study time point. This would have exposed our participants to radiation and was thus not included in the study protocol. Further, in the additional reference # 44 (Luoma K, Vehmas T, Riihimäki H, Raininko R. Disc height and signal intensity of the nucleus pulposus on magnetic resonance imaging as indicators of lumbar disc degeneration. Spine. 2001;26:680-686. doi 10.1097/00007632-200103150-00026), the validity of absolute disc height, alone or combined with other measures, as an indicator of early disc degeneration is questioned. 

We have continued to analyze our MRI data using the Pfirrmann classification (Pfirrmann CWA, Metzdorf A, Zanetti M, Hodler J, Boos N. Magnetic resonance classification of lumbar intervertebral disc degeneration. Spine. 2001;26:1873-1878. doi 10.1097/00007632-200109010-00011) which considers not only the signal intensity of the nucleus pulposus but also the disc height. In the present manuscript, we report the results based on the original study design with assessment of disc signal intensity using the Schneiderman classification. We are currently analyzing the results using the Pfirrmann classification.

Author action:

Lack of information about the disc height has been added to the Discussion as a further limitation of our study (Discussion, 10th and especially 11th chapter): Only T2-weighted sagittal images were obtained to reduce the scanning time in our young participants. Thus, our analysis was limited to changes in the SI of the intervertebral discs, which might have caused us to overlook other morphological changes related to LBP. For example, a more significant association between disc bulge and LBP has been suggested in younger adults compared to older subjects [19]. DD, however, is the most common structural abnormality in the pediatric spine on MRI [25] and thus of special interest clinically. 

For a clinically meaningful analysis of the evolution of disc height we would have needed a standing lumbar spine x-ray at each of the study time points. This would have exposed our participants to unnecessary radiation and was not included in the study design. Moreover, the validity of absolute disc height, alone or in combination with other measures, as an indicator of early disc degeneration has been questioned [44]. Pfirrmann et al., in their MRI investigation of 70 asymptomatic adults, concluded that the association of disc degeneration and disc height was stronger in older individuals compared to younger subjects [45]. 

Reviewer comment:

The authors chose to ignore the potential influence of BMI on SI even after the reviewer asked for clarification.

Author response:

Our initial plan for publication of the results was to first concentrate on the association between changes in disc signal intensity and LBP and then elaborate on the association of e.g., BMI and other demographics with disc signal intensity and LBP. In this re-revised manuscript, we have now included data on the influence of BMI on the disc to CSF -SI ratio. 

Author action: 

We have addressed the influence of BMI on the computerized quantitative SI (i.e. the ratio between disc and CSF SI) as follows (In Results, under “Disc to cerebrospinal fluid -SI ratio and its association to LBP”): “At Y19, BMI showed a statistically significant association to the disc to CSF -SI ratio at the L5/S1 level (p=0.035); no other statistically significant associations emerged at any of the disc levels at the different study time points (Table 3).” The data on BMI is illustrated in the new Table 3.

Table 3. The Spearman correlation between BMI and Disc to CSF -SI ratio 

 Y8

(N=92)

r (95% CI) Y12

(N=81)

r (95% CI) Y19

(N=71)

r (95% CI)

L3/L4 0.16 (-0.05 to 0.28) -0.03(-0.25 to 0.19) 0.05 (-0.19 to 0.28)

L4/L5 -0.00 (-0.21 to 0.21) -0.02 (-0.24 to 0.20) -0.04 (-0.27 to 0.20)

L5/S1 -0.14 (-0.33 to 0.07) -0.06 (-0.27 to 0.17) -0.25* (-0.46 to -0.02)

 p=0.035

 In the Discussion, we have included the following chapter to compare our results with previous literature: For the disc to CSF -SI ratio, higher BMI at Y19 correlated statistically significantly with lower relative SI at the L5/S1 level, although the correlation was only fair. High BMI at 16 years of age has previously been shown to be associated with lumbar DD among young males [46]. In another population-based cross-sectional study, overweight or obese adolescents and young adults had more severe DD than underweight or normal-weight individuals [23]. Compared to these previous findings, our results are more in line with those of van den Heuvel et al. who found no association between increased BMI and disc SI in their 9-year-old subjects [47]. 

Reviewer comment:

“The only significant finding was that the 4 participants with a dark disc at L4/L5 level reported LBP.” This statement is not clearer because the comparison is not stated. There are three ages and three types of intensity. What is compared to what to draw this conclusion?

Author response: 

We have further analyzed the data presented in Table 1. To avoid the problems of multiple testing, we performed the analysis using the most degenerated disc regardless of intervertebral level. After this analysis the previous statistically borderline significant (p=0.048) association between a dark disc at the L4/L5 level and LBP in four study subjects disappeared.

Author action:

We have made the necessary changes to this manuscript (please see above the discussion under statistical significance in the Tables).

Reviewer comment:

From Response: “We have added the results of our analysis of the relative change in height to the results section. The results confirm that by age 12 girls had already entered the growth spurt while with boys more growth occurred after the age of 12 years.” From manuscript: lines 203-209 show that the relative growth between males and females was similar and they were not compared statistically.

Author response: Thank you for this comment. In the revised manuscript, we only reported the confidence intervals.

Author action: We have added the following statement to the Results (2nd chapter): Between Y12 and Y19 the relative growth of males was significantly more than that of females (p<0.001).

Reviewer comment:

Separately, how can a growth of 19 cm have a p>0.001? It is likely less than 0.001.

Author response: We apologize for this unfortunate typo and thank the reviewer for pointing it out.

Author action: We have corrected the p-value to p<0.001.

Reviewer 2: (No Response)

Reviewer 3: Authors conduct a longitudinal cohort study to determine the history of disc changes from age 8 to 19 years old and evaluate the association between these changes and self-reported low back pain (LBP). They analyzed 208 participants and observe 54% prevalence of LBP at year 19. Also they observed the association between dark disc at L4/L5 level at year 19 and LBP.

Reviewer comment:

1. Abstract/line 224. “all four participants with a dark disc at…” only four participants in this analysis? The sample size is too small. Please comment on the generalizability of this result. Also, what statistical test was performed here for sample size of 4? Is it suitable for such a small sample?

Author response: 

As a response to the comment of Reviewer 1 regarding the association of disc signal intensity changes and LBP, we did further statistical analysis based on data in Table 2. To avoid the problems of multiple testing, we analyzed the data anew according to the most degenerated disc regardless of intervertebral level. With this analysis the previous statistically borderline significant (p=0.048) association between a dark disc at the L4/L5 level and LBP in four study subjects disappeared. 

Author action:

We have deleted all information on the above-mentioned finding from the abstract and the text proper. Specifically, in the abstract, the conclusion now states: Changes in disc SI were not associated with the presence of LBP in childhood, adolescence, or early adulthood. In the Results, we deleted the following statement: “When the disc levels were considered separately, the only significant finding was that all participants with a Dark disc at L4/L5 level at Y19 (n=4) reported LBP (p=0.048).” In the Discussion, the following statement was deleted: “The only statistically significant association between disc degeneration and LBP emerged in the four participants with a dark disc at L4/L5 level at the age of 19 years who all reported LBP.” In the revised manuscript, we stated “The most significant increase in both LBP and disc changes occurred after the pubertal growth spurt. The only statistically significant association between disc degeneration and LBP emerged in the four participants with a dark disc at L4/L5 level at the age of 19 years who all reported LBP.” This statement has now been modified as follows (Discussion, 1st chapter): The disc SI changes did not associate with the presence of LBP in childhood, adolescence or early adulthood. In the Conclusions, we deleted the following statement: “The only significant relationship between LBP and disc SI changes was noticed in the four participants with a dark disc at L4/L5 level at the age of 19 years.” The Conclusions now state: In this small study population, these disc SI changes did not have an association with LBP. Based on the abovementioned analysis, we have significantly simplified Table 2.

Reviewer comment:

Please clearly report the power analysis results so that readers know how to evaluate these insignificant results.

Author response: 

We did not perform a formal power analysis. When we started our study, lumbar spine MRI was still relatively rare and thus significantly more expensive than today. Financial constraints (funding) limited the number of participants to approximately 100, i.e. 300 MRI investigations. The last chapter of our manuscript highlighted this issue.

Author action: 

We have stated more clearly that this is a small study population. In Discussion, this has been emphasized as follows (last chapter): Thus, it is entirely possible that due to a small number of participants our study did not have enough power to detect possible associations between disc SI changes and LBP. Further, the Conclusions now emphasize: In this small study population, these disc SI changes did not have an association with LBP.

Yours sincerely,

Teija Lund, MD, PhD

---

## [Decision Letter · Decision Letter 2]

14 Sep 2022

The intervertebral disc during growth: signal intensity changes on magnetic resonance imaging and their relevance to low back pain

PONE-D-22-02909R2

Dear Dr. Lund,

We’re pleased to inform you that your manuscript has been judged scientifically suitable for publication and will be formally accepted for publication once it meets all outstanding technical requirements.

Kind regards,

Alejandro A. Espinoza Orías, PhD

Academic Editor

PLOS ONE

Additional Editor Comments (optional):

Reviewers' comments:

Reviewer's Responses to Questions

**Comments to the Author**

1. If the authors have adequately addressed your comments raised in a previous round of review and you feel that this manuscript is now acceptable for publication, you may indicate that here to bypass the “Comments to the Author” section, enter your conflict of interest statement in the “Confidential to Editor” section, and submit your "Accept" recommendation.

Reviewer #3: (No Response)

2. Is the manuscript technically sound, and do the data support the conclusions?

Reviewer #3: (No Response)

3. Has the statistical analysis been performed appropriately and rigorously? 

Reviewer #3: (No Response)

4. Have the authors made all data underlying the findings in their manuscript fully available?

Reviewer #3: (No Response)

5. Is the manuscript presented in an intelligible fashion and written in standard English?

Reviewer #3: (No Response)

6. Review Comments to the Author

Reviewer #3: (No Response)

7. PLOS authors have the option to publish the peer review history of their article (what does this mean?). If published, this will include your full peer review and any attached files.

Reviewer #3: No

---

## [Editor Report · Acceptance letter]

20 Sep 2022

PONE-D-22-02909R2 

The intervertebral disc during growth: signal intensity changes on magnetic resonance imaging and their relevance to low back pain 

Dear Dr. Lund:

I'm pleased to inform you that your manuscript has been deemed suitable for publication in PLOS ONE. Congratulations! Your manuscript is now with our production department. 

Kind regards, 

on behalf of

Dr. Alejandro A. Espinoza Orías 

Academic Editor

PLOS ONE